# Coupling Analysis of Urban Land Use Benefits: A Case Study of Xiamen City

**Xuanming Ji [1,2]**, **Kun Wang [1]**, **Tao Ji [3]**, **Yihua Zhang [1]** and **Kun Wang [2,4,\*]**

[1]   School of Finance and economics, Jimei University, Xiamen 361021, China; jxm@jmu.edu.cn (X.J); 201811202004@jmu.edu.cn (K.W.); yhzhang@jmu.edu.cn (Y.Z.)
[2]   The Institute of Financial Studies, Sichuan University, Chengdu 610064, China
[3]   Department of Computer Sciences, University of Wisconsin-Madison, Madison, WI 53706, USA; taoji@cs.wisc.edu
[4]   Business School, Sichuan University, Chengdu 610064, China
[\*]   Correspondence: liam_wang@stu.scu.edu.cn; Tel.: +86-17716353638

**Abstract:** The high coupling coordination of urban land use benefits is a significant factor for urbanization and sustainable urban development. This study, based on the statistical data from 2002 to 2017 of Xiamen City, constructs an index system that includes social, economic, ecological, and environmental benefits by evaluating the overall coupling coordination degree of land use benefits, using the entropy weight method (EWM), the coupling coordination degree (CCD) model, and the dynamic coupling coordination degree (DCCD) model. The results show that the coupling degree of Xiamen City's land use is relatively low, while showing a positive trend of development. In terms of the management of land use, the market should play a major role to achieve more efficient land use and promote industrial upgrading. The government should take responsibility for supplying infrastructure, perfecting related laws and regulations, intervening the land use according to the law of markets, and expanding the investment in education, as well as science and technology.

**Keywords:** land use benefits; entropy weight method; coupling coordination degree; Xiamen City

## 1. Introduction

According to the United Nations, approximately 70% of the population will be living in urban areas by 2050 [1]. China has made enormous progress in urbanization. Since the Chinese economic reform, the rate of urbanization has risen from 17.90% in 1978 to 58.20% in 2017, with a 1.04% annual growth rate, marking 2.5% higher than the global average [2]. New problems have resulted from the overly rapid urbanization, such as the deterioration in ecology, land degradation, human-land conflict, a decrease in overall urban land use benefits, and even the abandoning of arable land in the rural areas [3,4]. To this end, China has been strengthening land use management since the early 1990s, mainly protecting the arable land resources and limiting the growth of developed areas [5]. Land for urban construction, however, is a key element for socioeconomic development [6]. China's land for construction is still increasing with the rapid urbanization [7]. How to strike a balance among land uses, i.e., expand the benefits of land use while achieving healthy development, is a difficult problem that exists in the process of China's urbanization.

Land use benefits refer to the sum of all benefits in a city obtained from land arrangement, utilization, and optimization in terms of quantity and quality [8]. Ewing believed that, although urban expansion promotes the economic benefit of urban land use, it can lead to an increase in social and ecological environment costs, such as longer commuting times, waste of resources, and damage of the ecological system [9]. Fulton tested the significant relation between urban land expansion and

the economic benefit of urban land [10]. Ely and Morehouse believed that wealth production and distribution, as well as the protection of ecological resources, relied on the social benefit of land use [11]. Land use benefits include economic and social benefits, and also ecological and environmental benefits, and represents the sum of the four benefits [12]. The coupling degree of land use benefits is a crucial criterion in the evaluation of whether land use is reasonable as well as an important area in the research on sustainable land use [13,14]. Therefore, it is necessary that the sustainability of land use be studied leveraging the coupling relationships among land use benefits.

Xiamen is the most economically developed city in Fujian Province, situated on merely 1.37% of the land in the province, with 10.5% of the province's population and 13.52% of the province's gross product. Such a situation has incurred huge pressure on the ecological environment of the land. Thus, it is significant for the sustainable development of Xiamen City to study the coupling relationships of its land use benefits.

Given the above theoretical and realistic grounds, this paper, based on data between 2002 and 2017, takes Xiamen City as the research area and builds an index system of all four social, economic, ecological, and environmental benefits, using 21 indices, to evaluate the overall land use benefits. This paper also leverages the entropy weight method (EWM) to calculate the index weights, computes the land use benefits, and derives the coupling coordination degree (CCD) of the socioeconomic and ecological environment benefits of the land use of Xiamen City using the coupling coordination degree (CCD) and dynamic coupling coordination degree (DCCD) models, respectively. The contributions of this paper are as follows: (i) a more comprehensive index selection, leading to a more accurate measure of land use benefits; (ii) the use of the CCD and DCCD models, respectively, resulting in a more persuasive conclusion; and (iii) provision of an empirical basis for sustainable development and government policy making by calculating the coupling relationship of the land use in Xiamen City.

The remainder of this paper is organized as follows: Section 2 presents a review of related literature, Section 3 introduces the data processing and research methodology, Section 4 derives the results, Section 5 includes a discussion of the results, and Section 6 concludes the paper.

## 2. Literature Review

Since the main use of land is to benefit the economy, people have mainly focused on the economic benefit, and the law of rent in western economics has provided the theoretical foundation for evaluating the economic benefit of land use. Later, Ely and Morehouse discussed in Principles of Land Economics certain economic principles for land use, such as the principles of land scarcity, substitution, proportionality, and the principles of maximum reward of limiting factors [11]. They also emphasized that land use should meet social goals such as wealth production and fair distribution, as well as ecological environment protection [11]. They also believed that the government should utilize the means of economic leverage, political power, and legislation to improve healthy regional economic development, because individuals use land to maximize their personal benefits rather than the overall social benefits [15]. Dunn et al. proposed the concept of ecosystem services [16], while each researcher could have slightly different definitions of the ecosystem service functions. Daily defined the ecosystem service functions as the conditions and process to provide satisfaction to the natural ecological system and its process and he divided the functions into 15 categories [17]. Costanza et al. defined the ecosystem service functions as the benefits to the human group directly or indirectly from the ecosystem services, and categorizing them into 17 categories, and thereby derived the value of global ecosystem service functions at USD 33 trillion per year [18]. This result has been widely applied and the ecological benefit of land use has gained increasing attention. With the development of the sustainability theory, people have gradually become aware that the goal of land use is the sustainable utilization of land resources, which requires considering the economic, social, and ecological benefits. As a result, researchers have focused more on the research of the overall benefits of land use [19–21].

Nowadays, research on land use benefits is no longer confined to the evaluation of land use benefits, but also includes the relationships between land use benefits and other factors. Liu et al.

evaluated the input-output efficiency of China's urban land, using the data envelopment analysis model to analyze its coupling relationship with urbanization, and discovered that the input-output efficiency of China's urban land has low coupling with urbanization rate, spatially descends from east to west, and increases with the urban scale [22]. Wang et al., from a research with 27 cities in the Bohai Economic Rim as objects, built a coupling coordination model for land use benefits and urbanization, and discovered that certain spatial responsive relations existed between the coupling coordination and the region's economic development pattern [23]. Song et al. used the multilayer analytical method, coupling degree model, and coupling coordination degree model to discuss the law of mutual influence and interaction between urbanization and resource environment in Hubei Province [24]. Wang et al. leveraged general theories and methods in system science, taking Xianning City as an example, to build a theoretical model of the coupling coordination of socioeconomic and ecological environment benefits of intensive use of land, covering not only the socioeconomic and ecological environment benefits of intensive land use, but also their coupling and coupling coordination degrees [25]. Fan et al. introduced the theory of symbiosis to explain the target of the interaction between land use and industrial development, and therefore achieved improvement of land use benefits and industrial structure optimization, stepping towards the coordinated advancement of land use and industrial development. The symbiosis theory also provides the theoretical fundamentals and practical applications for land resource exploitation and industrial structure adjustment by studying the symbiotic relationship between land use and industrial development [26].

Generally speaking, there have been numerous researchers who have performed spatial and temporal studies on land use benefits and discussed their coupling relationships. From the perspective of time series, the research methodology is mainly to derive the land use benefits by constructing the evaluating index system of land use benefits, weighing the indices, and then calculating the land use benefits. Finally, the CCD model is leveraged to discuss the discrepancy of the time series of the land use benefits. Due to the difference in data obtainability and focuses, researchers have selected different indices with various weighing methods. For instance, Zhou et al. leveraged the analytic hierarchy process (AHP) to determine the weights of indices, which was simple and clear. The AHP has also provoked other researchers to think carefully about the relative significance of significances. However, unreasonable selections of factors, unclear implications, or incorrect relations among factors can compromise the quality of the AHP method, even leading to failed decisions [27]. Zhang et al. used the entropy weight method (EWM), which reflected the implications of the indices, and reached higher accuracy [28]. Zhu et al. combined the technique for order preference by similarity to ideal solution and the variation coefficient method to construct the index weights, which enabled multiplexing several indices from different levels of research into one single index, and also avoided subjectively assigning index weights. However, the index can lose its significance when its change is not large enough [29]. Liang et al. built a DCCD model based on the general system theory to study the coupling relationship of the land use benefits in Ningbo City, which provided a new perspective for studying the coupling relationships of land use benefits. Nevertheless, they used the Delphi method to calculate the index weights, and the result was relatively subjective [30]. Lin studied the benefits of using new urban land in Fujian Province with factor analysis, which effectively reduced the number of variable dimensions, preserving a large amount of original information with a smaller number of variables. However, this required high data accuracy. It was also difficult to discover data error during the data analysis in this approach and some special situations were prone to be ignored [31]. From a spatial perspective, spatial quantitative model, remote sensing, and geographical information system (GIS) have been utilized to analyze the spatial correlation and difference, as well as their cause, between regional overall land use benefits and the coupled development of the socioeconomic and ecological benefit systems, to theoretically provide fundamentals and scientific guides for inter-regional coordinated development [32–34]. A range of studies have covered the country, provinces, cities, counties, metropolitan regions, and metropolitan regions formed by groups of cities [35–41].

During the literature review, we found that although the index weighing methods varied, when studying individual regions, prior works mainly used the CCD and DCCD models independently, impacting the comprehensiveness of measure. Moreover, the selection of indices was relatively small, which failed to entirely reflect the land use benefits. Furthermore, the decision of many of the index weights was subjective. Taking into consideration the limitations of past research, this paper builds an index system that evaluates Xiamen's land use benefits from the aspects of social, economic, ecological, and environmental benefits. Then, the EWM, CCD, and DCCD models are applied to evaluate the overall coordination of the land use benefits in Xiamen, leading to a more realistic reflection on Xiamen's land use.

## 3. Data and Methodology

### 3.1. Region of Study

Xiamen City is located at 24°23′–24°54N and 117°53′–118°26′E in southeast Fujian. It is on the southeast coast of China beside Taiwan Strait and opposite from Taiwan and Penghu Islands across the strait (see Figure 1). Xiamen is a regional center with an international port and tourist attraction sites. It covers Xiamen Island, which is the center of the city, Gulang Island, Haicang Peninsula to the west, Jimei Peninsula to the north, Xiangan Peninsula to the east, and inland Tong'an District, as well as several smaller islands. Xiamen has 6 districts, spanning 1700.61 square kilometers of land, including tidal-flat areas, more than 390 square kilometers of sea, with a population of 411 million. The terrain of the city mainly consists of coastal plains, tables, and hills. The weather is mostly warm and rainy as the subtropical monsoon climate. As one of the four earliest special economic zones, Xiamen has gained huge progress in socioeconomic development under the support of China's policy. The standard of living of residents has drastically improved. The per capita housing area increased from 18.31 m$^2$ in 2002 to 30.85 m$^2$ in 2017, during which period, the Engel's coefficient decreased from 41.1% to 31.2%, and the GDP increased from 28,752 to 109,753 Chinese Yuan (calculated based on permanent resident), placing at the top in Fujian Province. Health care and education has obtained steady progress, for which the fiscal expenditure increased from CNY 124 million in 2013 to 1.772 billion in 2017, having multiplied by nearly 15 times. Xiamen is one of the fastest growing cities with the highest land use efficiency in China, contributing to 13.52% of the gross regional product of Fujian Province, while taking merely 1.4% of the land area.

For the past decades, with urbanization advancing and the economy developing rapidly, the urban area has been swallowing huge amounts of land for agricultural and environmental uses. From 2002 to 2017, the developed urban area expanded by three times from 94.27 square kilometers to 381.97 square kilometers. The arable land shrank by 20%, from 235.85 square kilometers to 188.76 square kilometers. The garden/forest land shrank by 727.19 square kilometers to 666.51 square kilometers. The status of land use in Xiamen City in 2017 is shown in Figure 2. The increase in urban construction land and industrialization has incurred enormous pressure on the ecological environment of the land. Pollutants such as industrial wastewater and waste gas have been depravating the quality of ecological environment, lowering the bearing capacity of the land, which can seriously hinder Xiamen's sustainable development. Therefore, it is extremely necessary to coordinate the land use benefits, and therefore improve the efficiency of land use and facilitate the city's development.

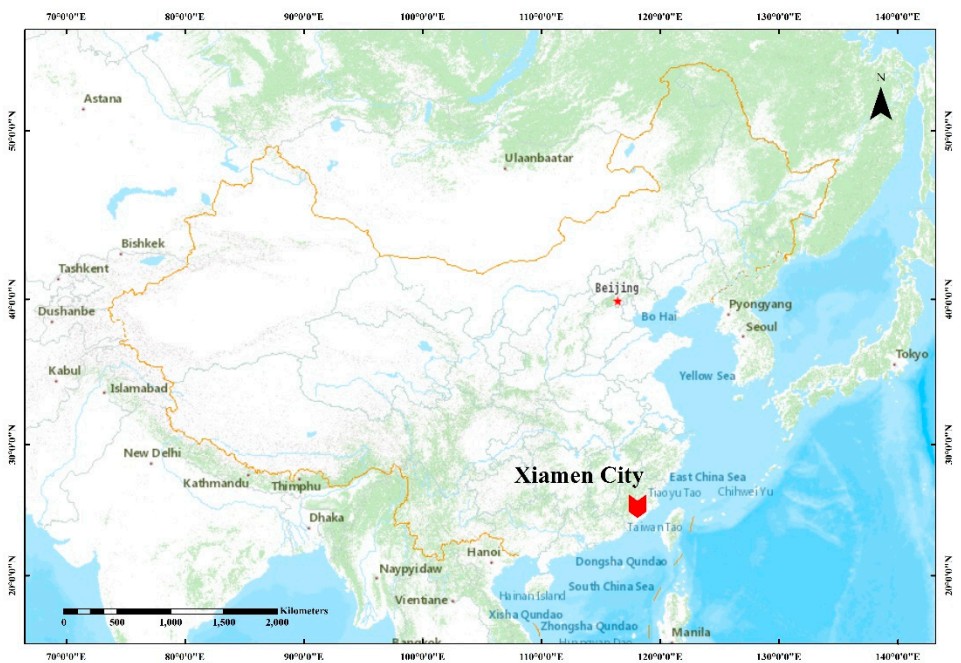

**Figure 1.** The location of study area.

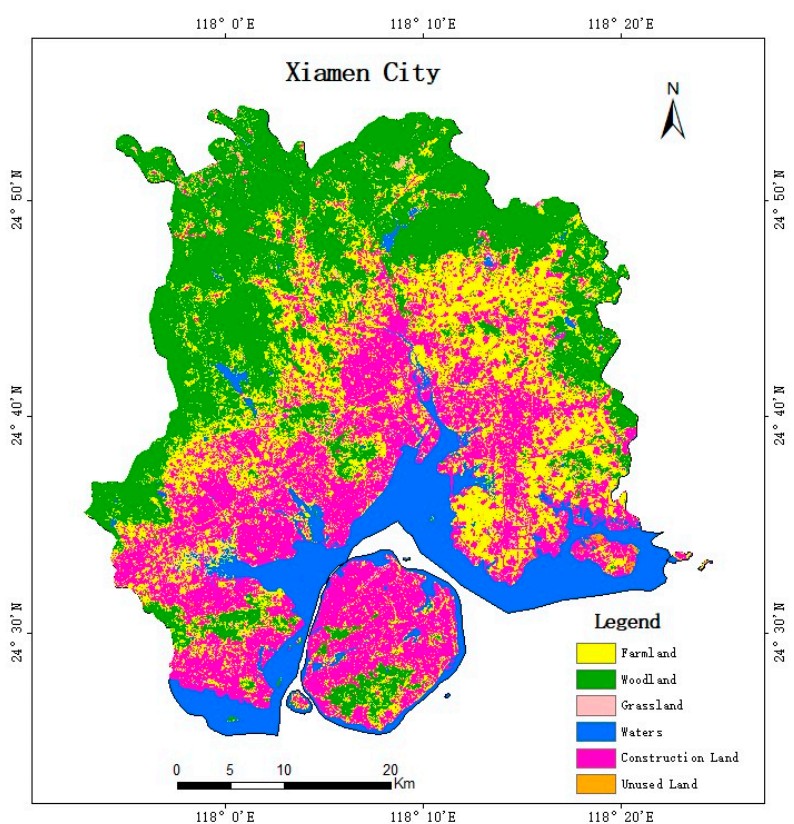

**Figure 2.** Land use status of Xiamen in 2017.

### 3.2. Data Source and Processing

The data used in this study was mainly obtained from the Yearbook of Xiamen Special Economic Zone 2013–2018, the Fujian Statistical Yearbook 2003–2018, and the China City Statistical Yearbook 2013–2018. Missing data is supplemented with that from local government announcements. Note that

a yearbook in China records the data of the previous year. For example, the Fujian Statistical Yearbook 2003–2018 records the data from 2002 to 2017.

To mitigate the impact of different dimensions or orders of magnitude of indices, first, we performed dimensionless processing to the data. We divide the indices into two categories, positive index and negative index, according to their effect on the system, and standardized the data by the following:

$$Y_{ij} = \begin{cases} (X_{ij} - m_i)/(M_i - m_i) & (Positive\ indicator) \\ (M_i - X_{ij})/(M_i - m_i) & (Negative\ indicator) \end{cases} \tag{1}$$

where $X_{ij}$ is the value and $Y_{ij}$ is the standardized value; $i$ is the number of index, ranging from 1 to 21; and $j$ is the year, ranging from 1–16, representing 2002–2017, respectively. Negative indices are population density in developed areas, Engel's coefficient of the residents, urban expansion rate (UER), industrial solid waste per unit area, industrial waste gas per unit area, and industrial wastewater per unit area in developed areas, all the rest being positive indices. Thus, all indices fall in the interval [0,1] after standardization.

### 3.3. Building the Index System of Land Use Benefits

We built an index system for evaluating the land use benefit from four aspects, i.e., social, economic, ecological, and environmental benefits, since the land use benefit is a composition of these four benefits [12]. Tian et al. believed that the social benefit of land use was mainly reflected by the change in the environment of the developed urban area, as well as the gathering and dispersion of urban population [42]. Therefore, they chose the population density of a developed area to represent the change in the urban land carrying population; and they used the per capita housing space to reflect the change in urban residents' living standards [42]. They also leveraged a proportion of the land for traffic use in the developed area to represent the development of traffic facilities and a proportion of land for public facilities to represent the change in public facilities [42]. Liang et al. pointed out that the social benefit should also include the wealthiness of the residents, as well as the employment and education conditions [30]. They used the Engel's coefficient to reflect the wealthiness, the urban employment rate to represent the employment conditions, and utilized the proportion of the population with post-secondary degrees to represent the education condition [30]. Due to a lack of data, we used the per-10K full-time teacher to represent the education condition in Xiamen.

The economic benefit of land use is mainly reflected by the urban gross product and fiscal revenue in the process of land use [43]. Due to the difference in the industrial structures across different cities, we divided the urban gross product into three parts by industry and utilized the increased value of each industry per unit area, as well as the fiscal revenue per unit area, to measure the economic benefit of land use.

Wang and Guo et al. measured the ecological benefit with green coverage in a developed area, proportion of garden/forest land, forest coverage, and per-capita green area [44,45]. Because of a change in statistical methods related to the green area in Xiamen, during the period of our study, we excluded the per-capita green area from our system. Meanwhile, the interval between data points of forest coverage was relatively long. Thus, we also excluded it from our indices. Complementarily, we used the urban expansion rate to indirectly represent the influence of urban expansion on the ecological environment.

As for the environmental benefits, many research works have used indices such as the overall reuse rate of industrial solid waste, proportion of concentrated and processed water, and proportion of harmless treatment of household garbage, while ignoring the indicators of pollution [46]. We added the industrial wastewater per unit developed area, industrial waste gas per unit area, industrial solid waste per unit developed area, and sewage per unit developed area [47].

Eventually, we used four primary indices, which were social, economic, ecological, and environmental benefits, as well as 21 secondary indices, as shown in Table 1:

**Table 1.** The evaluation index system of land use benefits.

| Item | Primary Index | Secondary Index | Weight |
|---|---|---|---|
| Socioeconomic benefits | Social benefits | Population density of developed area (per km$^2$) | 0.0204 |
| | | Urban residents per capita use area (per m$^2$) | 0.0363 |
| | | Urban employment rate (%) | 0.0392 |
| | | Full-time teacher per ten thousand permanent residents (per 10,000) | 0.1241 |
| | | Engel's coefficient of residents (%) | 0.0463 |
| | | Land for traffic use in developed area (%) | 0.1239 |
| | | Land for public facility in developed area (%) | 0.0422 |
| | Economic benefits | Fiscal revenue per unit area (CNY 10,000) | 0.0661 |
| | | Increase in the tertiary sector product per unit area (CNY 10,000) | 0.0692 |
| | | Increase in the secondary sector product per unit area (CNY 10,000) | 0.0513 |
| | | Increase in the primary sector product per unit area (CNY 10,000) | 0.0420 |
| Ecological environment benefits | Ecological benefits | Green coverage in developed area (%) | 0.0370 |
| | | Urban expansion rate (%) | 0.0158 |
| | | Proportion of garden/forest land (%) | 0.0781 |
| | Environmental benefits | Industrial wastewater per unit developed area ($10^4$ t/km$^2$) | 0.0260 |
| | | Industrial waste gas per unit developed area ($10^8$ $m^3$/km$^2$) | 0.0228 |
| | | Industrial solid waste per unit developed area ($10^4$ t/km$^2$) | 0.0411 |
| | | Overall reuse rate of industrial solid waste (%) | 0.0284 |
| | | Proportion of concentrated and processed water (%) | 0.0169 |
| | | Sewage per unit developed area ($m^3$/km$^2$) | 0.0199 |
| | | Proportion of harmless treatment of household garbage (%) | 0.0530 |

*3.4. Index Weighing*

By analyzing the index weighing approaches of related works, we discovered that existing works mostly utilize multilayer analysis and the Delphi method, which are relatively subjective and can compromise the accuracy of the results. To better avoid subjective influence and reflect the information in the data more objectively, we used an objective weighing method. Li et al., after comparing the entropy weight, mean-variance, and range methods, discovered that the entropy weight method was applicable to any data other than standardized data [48]. They also pointed out that, although the entropy weight method effectively reflected the difference between indices, it did not reflect the difference between any two evaluated objects [48]. Such a defect, however, to us is negligible. Therefore, we leveraged the entropy weight method as the weighing method of our index system.

The steps are as follows [28]:

1. Calculate the proportion of the *j*-th year of the *i*-th index as

$$P_{ij} = Y_{ij} / \sum_{j=1}^{m} Y_{ij} \tag{2}$$

where *m* is the maximum year, which is 16, in our context;

2. Calculate the entropy of the *i*-th index as

$$E_i = -\frac{1}{\ln m} \sum_{j=1}^{m} P_{ij} \ln P_{ij} \tag{3}$$

when $P_{ij} = 0$, we define $\lim\limits_{P_{ij} \to 0} P_{ij} \ln P_{ij} = 0$;

3. Calculate the weight of the $i$-th index as

$$w_i = \frac{1 - E_i}{\sum_{i=1}^{n}(1 - E_i)}. \tag{4}$$

where $n$ is the number of indices, which is 21, in our context.

### 3.5. Coupling Coordination Degree (CCD) Model

Coupling is a concept in physics, referring to the interplay of two or more systems or form of exercise where they affect each other [49]. A coupling degree is usually leveraged to measure the extent of the interplay among such systems or motion. However, the coupling degree does not reflect the level of development of each individual system, leading to a phenomenon where each individual system has a low level of development while having a high coupling degree [50]. The coordination degree, however, measures the extent of harmony and compatibility in the development of systems. It reflects the coordination condition, and also whether the systems are promoting or constraining each other. Liao et al., first, combined the coupling degree and coordination degree and proposed the coupling coordination degree (CCD) model to measure the CCD of a system or among multiple systems, and provided the intervals of CCD [51]. Since then, the CCD model has been widely used in system evaluation.

We built the CCD model using the following steps [52]:

1. Calculate the overall benefit of the socioeconomic and ecological environment systems

$$f_j(x) = \sum_{i=1}^{n} a_i x_{ij}, \tag{5}$$

$$g_j(y) = \sum_{i=1}^{m} b_i y_{ij}, \tag{6}$$

where $f_j(x)$ and $g_j(y)$ stand for the socioeconomic and ecological environment benefits of land use in the j-th year; $x_{ij}$ and $y_{ij}$ represent the standard value of the i-th index in the j-th year in the two systems, respectively; and $a_i$ and $b_i$ represent the weight of the corresponding index.

2. Calculate the CCD of the socioeconomic and the ecological environment systems:

$$C_j = 2\sqrt{\frac{f_j(x) \times g_j(y)}{\left(f_j(x) + g_j(y)\right)^2}}, \tag{7}$$

$$T_j = \alpha f_j(x) + \beta g_j(y), \tag{8}$$

$$D_j(x,y) = \sqrt{C_j \times T_j}, \tag{9}$$

where $C_j$ represents the coupling degree of the socioeconomic and ecological environment benefits in the j-th year and $C_j \in [0,1]$ ; $T_j$ stands for the overall coordination degree of the socioeconomic and ecological environment benefits of land use in the j-th year; $D_j$ is the coupling coordination degree of the two benefits in the j-th year and $D_j \in [0,1]$; $\alpha$ and $\beta$ are the contributions of the socioeconomic and ecological environment systems, respectively. In the particular case of Xiamen City, we determined the values of $\alpha$ and $\beta$ according to the actual calculation result of the socioeconomic and ecological environment benefits and $\alpha = 0.66$ and $\beta = 0.34$. According to the CCD value D, considering the existing research works, we divided the CCD into 3 stage and 10 categories [42], as shown in Table 2.

The CCD model inherits the advantages of the coupling and coordination degrees, being able to represent both the mutual influence and the coordination conditions between systems. However,

we discovered, by observing the equations, that the calculation of CCD mainly relied on the values of the two systems at some point in time, which ignored the dynamic changes of the systems. The combination of changes in the socioeconomic and ecological environment benefits can vary. For instance, the former can increase rapidly while the latter decreases, or both can remain unchanged. Such dynamic changes also reflect the mutual influence between systems, while left ignored by the model. To make up for such a defect, we proposed a dynamic coupling coordination degree model as follows:

**Table 2.** Discriminating standards of the coupling coordination degree.

| Stage | D Value | Category |
|---|---|---|
| Uncoordinated development | 0~0.09 | Extremely uncoordinated |
| | 0.10~0.19 | Seriously uncoordinated |
| | 0.20~0.29 | Moderately uncoordinated |
| | 0.30~0.39 | Slightly uncoordinated |
| Transitional development | 0.40~0.49 | At the edge of being uncoordinated |
| | 0.50~0.59 | Barely coordinated |
| | 0.60~0.69 | Slightly coordinated |
| Coordinated development | 0.70~0.79 | Moderately coordinated |
| | 0.80~0.89 | Well coordinated |
| | 0.90~1.00 | Perfectly coordinated |

### 3.6. Dynamic Coupling Coordination Degree (DCCD) Model

Li and Ding first proposed the coordination mechanism of sustainable development, having not only applied it to the coordinated evaluation of the socioeconomic system and resource environment systems, but also divided the coupling degree of complex systems into four stages [53]. On the basis of their idea and considering the existing works, we built the DCCD model. Since both the socioeconomic and the ecological environment systems are nonlinear [43], we can rewrite the evolution equation as:

$$\frac{\mathrm{d}x(t)}{\mathrm{d}t} = f(x_1, x_2, \ldots, x_n) \ i = 1, 2, \ldots, n \tag{10}$$

where $f$ is a nonlinear function with respect to $x_i$, and $x_i$ is a function with respect to $t$ which can also be written as $x_i(t)$. We write it as $x_i$ below for convenience. Similarly, $y_i$ is also a function with respect to $t$. According to Lyapunov's first approximation theorem, the motion stability of a nonlinear system depends on the properties of the eigenvalue of linear approximation system [54]. Thus, a linear approximation of a nonlinear system can be derived by applying the Taylor expansion to the equation at the origin and omitting the higher order items, while keeping the motion stability of the nonlinear system:

$$\frac{\mathrm{d}x(t)}{\mathrm{d}t} = \sum_{i=1}^{n} a_i x_i \ i = 1, 2, \ldots, n. \tag{11}$$

According to such a property, we construct the general function for socioeconomic and ecological environmental systems [55].

$$f(x) = \sum_{i=1}^{n} a_i x_i \ i = 1, 2, \ldots, n \tag{12}$$

$$g(y) = \sum_{i=1}^{m} b_i y_i \ i = 1, 2, \ldots, m \tag{13}$$

where $x_i$, $y_i$ are elements in each system (both are functions with respect to time) and ai, bi are weights of the element.

Since the socioeconomic and ecological environment benefits have tight relationships, being able to affect, constraint, and promote each other [42], we assume that a complex system of land use is formed out of the socioeconomic and ecological environment systems together. It is obvious that this

complex system has two major elements according to the general theory of systems [55]. The evolution equations of the complex system can be written as:

$$\begin{cases} A = \dfrac{\mathrm{d}f(x)}{\mathrm{d}t} = \alpha_1 f(x) + \beta_1 g(y) \\ B = \dfrac{\mathrm{d}g(y)}{\mathrm{d}t} = \alpha_2 f(x) + \beta_2 g(y) \end{cases} \tag{14}$$

A and B are the evolution state of the socioeconomic and ecological environment systems, respectively, under internal and external impacts. From Equation (14) one can see that A and B are linear combinations of $f(x)$ and $g(y)$. When A changes, $f(x)$ and $g(y)$ also change, which also affects B, and vice versa, and therefore A and B can affect each other. The change in any system results in changes in other systems, further leading to the change in the entire complex system.

$$v_A = \frac{\mathrm{d}A}{\mathrm{d}t} \tag{15}$$

$$v_B = \frac{\mathrm{d}B}{\mathrm{d}t} \tag{16}$$

where $v_A$ and $v_B$ represent the rate of evolution in the socioeconomic and ecological environment systems, respectively.

Since the complex system contains the socioeconomic and ecological environment systems only, when these two systems are coordinated, the complex system is also coordinated. The evolution rate of the complex system is affected by that of the two systems. Therefore, the complex system's evolution rate, $v$, can be expressed as a function with respect to $v_A$ and $v_B$, i.e., $v = f(v_A, v_B)$, from which we can study the coupling relationship of the complex system and its two components by controlling $v_A$ and $v_B$ and analyzing the change in $v$ [43].

Because the evolution rate of the socioeconomic benefit system fits the S-shape mechanism [56,57], we assume it changes periodically. Meanwhile, the evolution rate of the ecological environment benefit system also changes periodically due to the effects of the socioeconomic benefit system [58]. In each period, the change in $v$ is caused by $v_A$ and $v_B$, so we can analyze $v$ by projecting the evolution trace of $v_A$ and $v_B$ onto the two-dimensional plane formed by $(v_A, v_B)$. The eventual projection is an ellipse because the maximum evolution rate of the socioeconomic system is greater than that of the ecological environment system in extreme conditions [53], as shown in Figure 3. Variable $\alpha$ represents the angle between $v_A$ and $v_B$, and is subject to:

$$\alpha = \arctan\frac{v_A}{v_B} \tag{17}$$

The DCCD of the complex system can be determined once $\alpha$ is settled. In one period, the complex system goes through four stages, i.e., low-level symbiosis (I), coordinated development (II), extreme development (III), regenerative development (IV) [59]. See Table 3 for detailed standard.

The DCCD model is based on the systems theory. It complements the CCD model in terms of the lack of system dynamics and provides a new perspective for measuring the mutual influence between systems. However, the DCCD model has some drawbacks. First, we leveraged Lyapunov's first approximation theorem to process the equations, replacing the nonlinear system with a linear system, ignoring the high-order (nonlinear) terms in the Taylor's expansion. This could have led to inaccuracy between the ideal and real states. Such inaccuracy is hard to quantify. Secondly, in the DCCD model, the coupling degree is only related to the rate of evolution of the system, while the state of the system is ignored. For example, when the ecological environment benefits are high and the socioeconomic benefits are low but rapidly growing, although the ecological environment benefits are shrinking, they are not constraining the socioeconomic benefits. Third, the model assumes self-development of the two systems, ignoring external influences, which can incur misleading conclusions. For instance, when the ecological environment benefit system reaches the limit, technological advancement can still accelerate

the evolution of the socioeconomic system. By comparing the features of the two models, we found that relying solely on one of them could cause misleading conclusions. Thus, we used two models to analyze the socioeconomic and ecological environment benefits, which could give better accuracy.

**Table 3.** The coupling degree of the land use benefits system.

| Stage | $\alpha$ | Stage of development | The Performance |
|---|---|---|---|
| I | $-90 < \alpha \leq 0°$ | Low-level symbiosis | The intensity of land use is relatively low. The socioeconomic benefit is insignificant. The impact on ecological environment is weak. |
| II | $0 < \alpha < 45$ | Primary coordinated development | $v_A < v_B$, The socioeconomic benefit grows slower than ecological environment benefit. The socioeconomic development begins to assert pressure on ecological environment. |
|  | $\alpha = 45.$ | Harmonic development | $v_A = v_B$, The socioeconomic benefit grows at the same rate as ecological environment benefit. Both systems are in harmony. |
|  | $45 < \alpha \leq 90.°$ | Co-development | $v_A > v_B$, The socioeconomic benefit grows faster than ecological environment benefit. Ecological environment starts to constrain socioeconomic development, though the conflict is not significant. |
| III | $90 < \alpha \leq 180$ | Extreme development | As ecological environment deteriorates, the socioeconomic benefit slips down from the top. The conflict between the two becomes increasingly serious, eventually leading to the collapse of the complex system. |
| IV | $-180 < \alpha \leq -90°$ | Regenerative development | Disintegration of the old system due to the bad status of socioeconomic and ecological environment benefits in the remaining land use. A new system comes into being. |

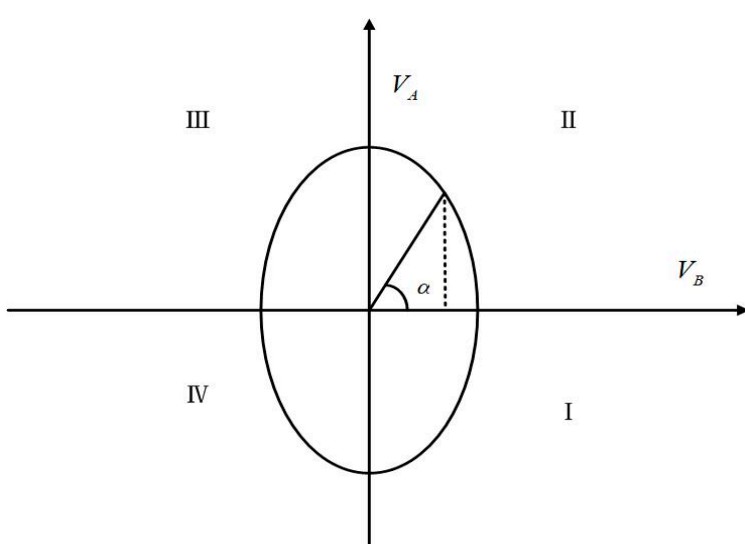

**Figure 3.** Process of coupling state of land use benefits system.

## 4. Results

*4.1. Time Series Analysis of the Socioeconomic and Ecological Environment Benefits of Land Use*

As shown in Table 4, the socioeconomic benefit fluctuated around 0.1 from 2002–2004, then around 0.15 after a growth to 0.1552 in 2005, until the steady growth from 2009–2017 to 0.5914, by a total of

0.4435, and an annual growth rate of 14.7%. From 2005–2009, the change in socioeconomic benefits slowed down, mainly due to the fact that the development of the city's infrastructure lagged behind, while the economy grew rapidly. As a result, the overall change in benefits was insignificant. The huge subprime lending crisis significantly impacted economic benefit. Nevertheless, with the establishment of the Western Taiwan Straits Economic Zone, the Western Taiwan Strait strategy being upgraded to a national strategy, and the cross-strait relationship developing towards peace, Xiamen narrowly escaped the economic crisis. After 2009, Xiamen's socioeconomic benefits grew rapidly. The ecological environment benefits of land use rose with fluctuations between 2002 and 2017, by a total of 0.0853. From 2002–2009, the ecological environment benefits were higher than the socioeconomic benefits and were increasing, reaching the highest in 2008, thanks to the transition of Xiamen's economic structure and emphasis on environmental protection. To prepare for the 2008 Olympic Games and to consolidate and deepen the results of obtaining the title of National Environmental Protection Exemplary City, Xiamen City enforced a plan to do so, outlining the division of responsibilities among each department of the city government and lower-level governments. Environmental protection was taken into the evaluation of the city mayor and the district governors. The ecological environment benefit peaked in 2008, then saw a brief decrease, and restored slow growth after 2011. Due to the 2011 Xiamen City Regulation on Power Saving, the new environmental protection regulation coming into force had positive effects on the public environment and indirectly influenced residents' environment-related behaviors [60]. After 2011, industrial wastewater, as well as gas and solid per unit area decreased annually and the ecological environment benefits increased but was still slower than the growth rate of socioeconomic benefits. Since then, the socioeconomic benefits of land use have been higher than the ecological environment benefits, with a widening gap. From 2002–2017, the overall benefit of land use was increasing, and the growth rate was also increasing, fitting a quadratic curve.

**Table 4.** The comprehensive evaluation values of the socioeconomic and ecological environment benefits of land use.

| Year | f(x) | g(y) | Total Benefit |
|------|------|------|---------------|
| 2002 | 0.1010 | 0.1502 | 0.2513 |
| 2003 | 0.0880 | 0.1858 | 0.2739 |
| 2004 | 0.1194 | 0.1515 | 0.2708 |
| 2005 | 0.1552 | 0.1572 | 0.3125 |
| 2006 | 0.1536 | 0.1967 | 0.3503 |
| 2007 | 0.1523 | 0.2209 | 0.3733 |
| 2008 | 0.1515 | 0.2263 | 0.3778 |
| 2009 | 0.1479 | 0.2112 | 0.3592 |
| 2010 | 0.2165 | 0.1726 | 0.3892 |
| 2011 | 0.2646 | 0.1640 | 0.4286 |
| 2012 | 0.2847 | 0.1913 | 0.4759 |
| 2013 | 0.3194 | 0.2012 | 0.5206 |
| 2014 | 0.4236 | 0.2037 | 0.6273 |
| 2015 | 0.5071 | 0.2277 | 0.7348 |
| 2016 | 0.5602 | 0.2001 | 0.7603 |
| 2017 | 0.5914 | 0.2355 | 0.8269 |

Table 4 shows the socioeconomic benefits and ecological environment benefits calculated according to Equations (12) and (13).

The following broken line statistical chart (see Figure 4) has been made based on Table 4.

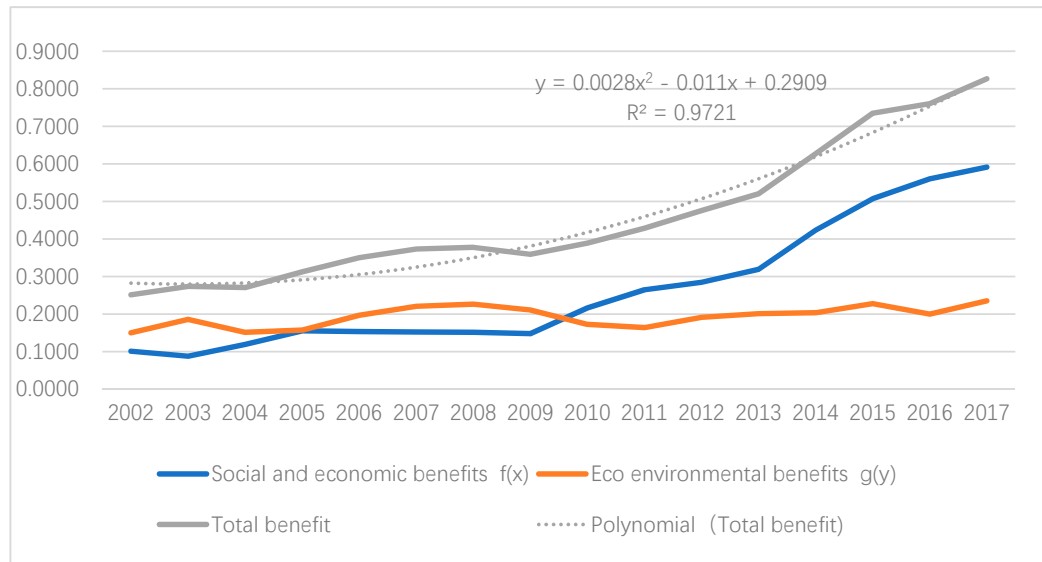

**Figure 4.** The comprehensive evaluation values of the socioeconomic and ecological environment benefits of land use.

*4.2. Time Analysis of the Coupling Coordination Degree (CCD) of the Socioeconomic and Ecological Environment Benefits of Land Use*

From Table 5, the coupling degree C of the socioeconomic and ecological environment benefits of land use lies between 0.8807 and 1, with a volatility of 13.5%, of which 93.8% is above 0.9. Therefore, this variable is relatively stable, implying a high level of correlation between the two benefits. The coupling coordinated degree (CCD) value D lies between 0.351 and 0.6109, with a volatility of 74%, which is relatively more significant, the reason being that, although the coupling degree reflects the synchronization, it does not imply the direction of their co-motion. The coupling coordination degree D, meanwhile, considers the extent of co-motion the two benefits. From the fourth column of Table 5, one can see that the socioeconomic and ecological environment benefits went through the following four stages between 2002 and 2017: slightly uncoordinated (2002–2005), at the edge of being uncoordinated (2006–2012), barely coordinated (2013–2016), and slighted coordinated (2017). We performed linear regression on D and the years. The result is:

$$D = 0.0168t + 0.3171 \quad R^2 = 0.944 \tag{18}$$

**Table 5.** The coupling coordination degree.

| Year | Coupling Degree C | Coupling Coordination Degree D | Category |
|------|-------------------|-------------------------------|----------|
| 2002 | 0.9806 | 0.3510 | Slightly uncoordinated |
| 2003 | 0.9341 | 0.3577 | Slightly uncoordinated |
| 2004 | 0.9930 | 0.3667 | Slightly uncoordinated |
| 2005 | 1.0000 | 0.3953 | Slightly uncoordinated |
| 2006 | 0.9924 | 0.4169 | At the edge of being uncoordinated |
| 2007 | 0.9830 | 0.4283 | At the edge of being uncoordinated |
| 2008 | 0.9802 | 0.4303 | At the edge of being uncoordinated |
| 2009 | 0.9844 | 0.4204 | At the edge of being uncoordinated |
| 2010 | 0.9936 | 0.4397 | At the edge of being uncoordinated |
| 2011 | 0.9721 | 0.4564 | At the edge of being uncoordinated |
| 2012 | 0.9806 | 0.4831 | At the edge of being uncoordinated |
| 2013 | 0.9739 | 0.5035 | Barely coordinated |
| 2014 | 0.9365 | 0.5420 | Barely coordinated |
| 2015 | 0.9249 | 0.5829 | Barely coordinated |
| 2016 | 0.8807 | 0.5786 | Barely coordinated |
| 2017 | 0.9027 | 0.6109 | Slightly coordinated |

From the equation, the CCD grows at a rate of 0.0168 annually, from which it is estimated that Xiamen can reach perfectly coordinated in 2034.

*4.3. Time Analysis of the Dynamic Coupling Coordination Degree (DCCD) of the Socioeconomic and Ecological Environment Benefits of Land Use*

By performing nonlinear regression on the above curves, we obtain:

$$\text{A} = 0.0029t^2 - 0.0163t + 0.1324 \ \text{R}^2 = 0.9781 \tag{19}$$

From Figure 5, fitting on B and j result in a low $\text{R}^2$, which means low accuracy. We select different time points as the boundary and observe $\text{R}^2$, then choose the point that optimizes fitting degree on both sides. Eventually, we perform regression on two intervals, i.e., 2002–2009 and 2010–2017. The results are shown in Figures 6 and 7.

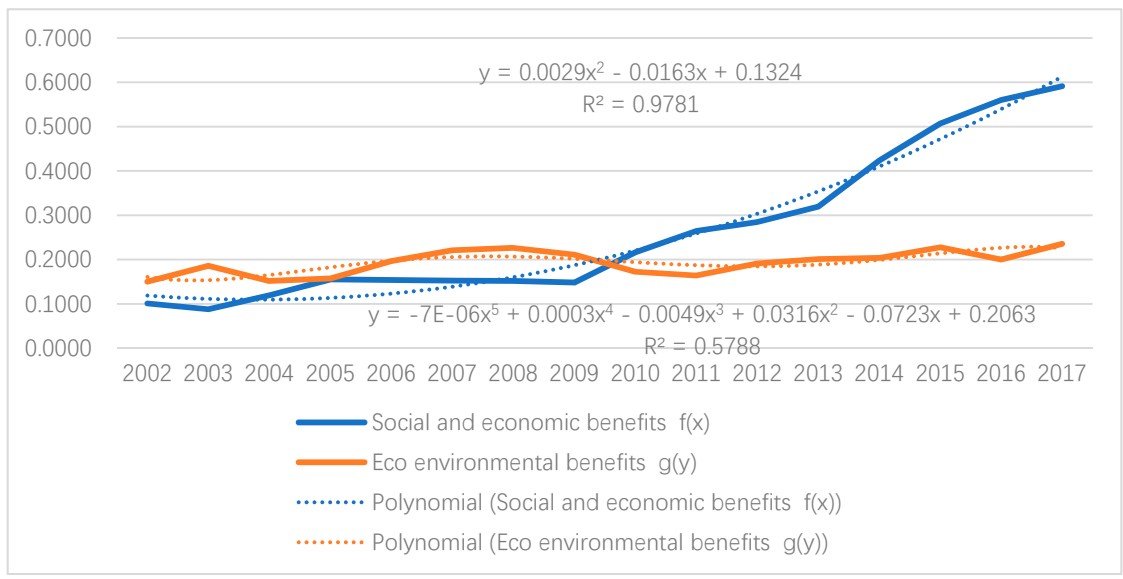

**Figure 5.** The comprehensive evaluation values of the socioeconomic and ecological environment benefits of land use.

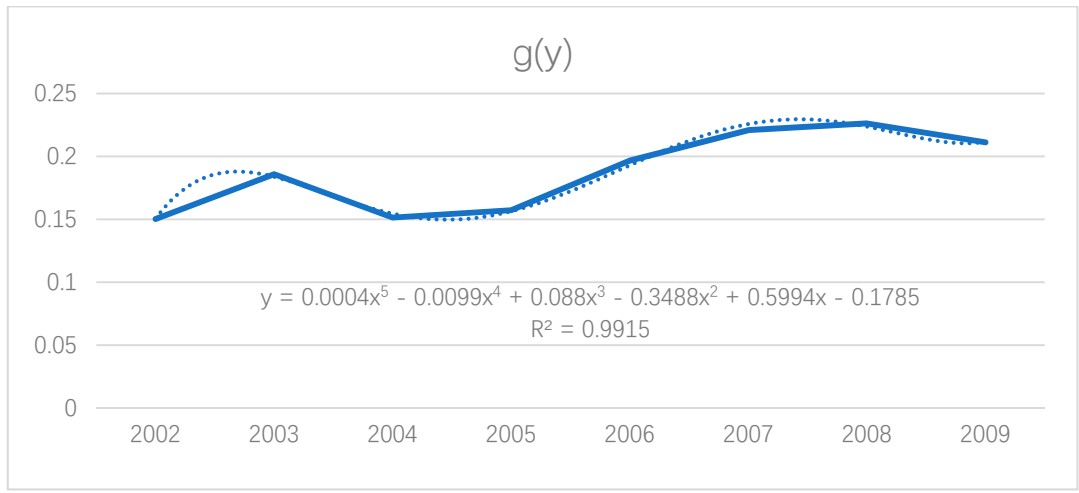

**Figure 6.** The comprehensive evaluation values of ecological environment benefits of land use.

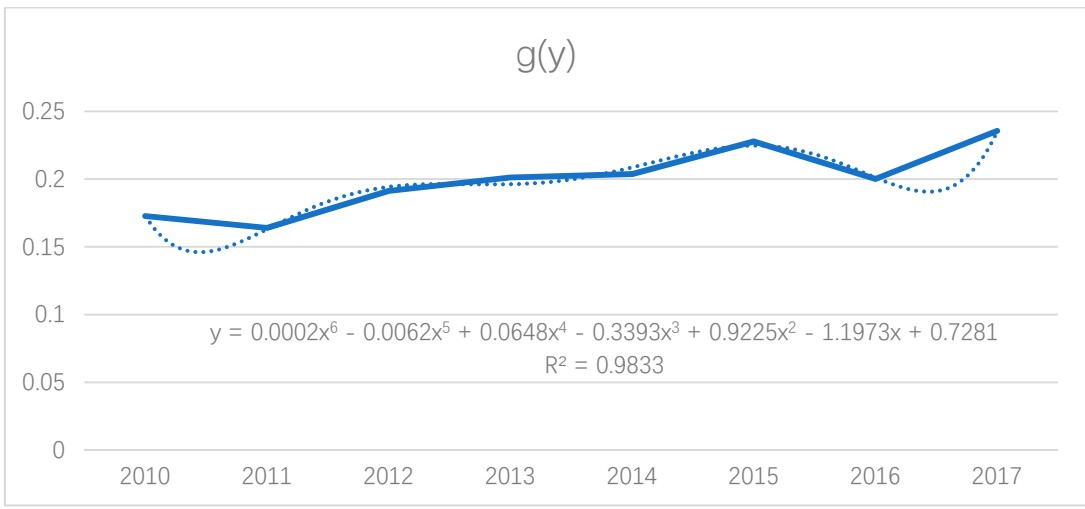

**Figure 7.** The comprehensive evaluation values of ecological environment benefits of land use.

From Equations (15) and (16), we obtain:

$$v_A = 0.0058t - 0.0163 \tag{20}$$

$$v_B = \begin{cases} 0.002t^4 - 0.0396t^3 + 0.264t^2 - 0.6976t + 0.5994 & 1 \le t \le 8 \\ 0.0012T^5 - 0.031T^4 + 0.2592T^3 - 1.0176T^2 + 1.845T - 1.1973 & 1 \le T \le 8 \end{cases} \tag{21}$$

where t ranges from 1 to 16, corresponding to the years 2002–2017, respectively, when $9 \le t \le 16$, $T = t - 8$. According to Equation (17), we get $\alpha$ and plot the evolution curve. The results are shown in Figure 8.

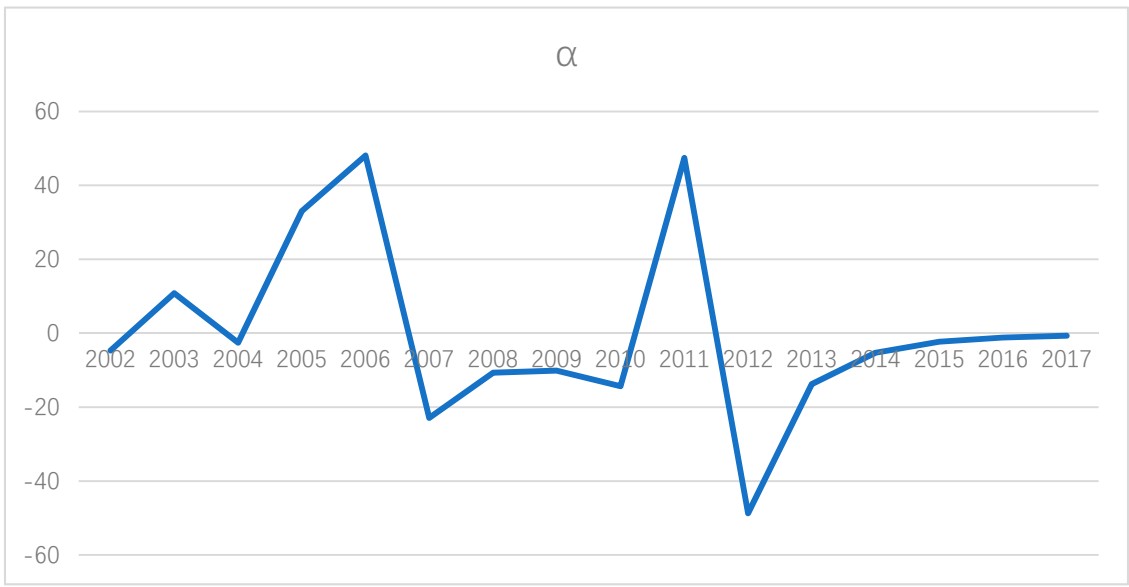

**Figure 8.** The dynamic coupling coordination degree of the socioeconomic and ecological environment benefits of land use.

From Figure 8, Xiamen was mainly in the stages of low-level symbiosis and primary coordinated development. The speedup of urbanization while the infrastructure construction was lagging behind, and the over concentration of population have contributed to the social benefit offsetting the economic benefit brought by economic development. The ecological environment benefits, however, were rising as the environment protection proceeded. Thus, the direction of change in socioeconomic and

ecological environment benefits is the same. The former grew faster than the latter, but the two growth rates were approaching each other. In 2006, the two growth rates reached synchronization, and the coordination attained the optimal. From 2007–2010, the progress of environment protection and the breakthrough in the controlling of industrial pollutants had some negative impacts on the economy. The industrial structure also changed, with the secondary sector surpassed by the tertiary sector. During this transitional period, the impact of socioeconomic development on ecological environment was weak. Under the impact of the subprime mortgage crisis in 2009, the development of the services industry was limited. From 2009–2012, the secondary industry developed rapidly, though at the cost of the ecological environment, containing the growth in ecological environment development. With the slowing growth of ecological benefit and the accelerating growth of the socioeconomic benefits, the optimal coordination was reached in 2011. Afterwards, since the growth rate of socioeconomic benefits was much higher than that of the ecological environment benefits, Xiamen was at the critical point between low-level symbiosis and primary coordinated development. This is because the economic growth was mainly in the tertiary industry, which produced less pollution. The treatment of the environment was also strengthened. Therefore, the ecological environment was not a constraint to socioeconomic development.

## 5. Discussion

By comparing the results from the CCD and DCCD models, we found that their outputs of the coupling coordination categories of Xiamen's land use benefits were consistent, both indicating a relatively low level with some distance to the optimal. However, the CCD model showed that the coupling coordination degree was growing linearly with gentle changes, while the DCCD model indicated that the degree was fluctuating and could change intensively under special circumstances, such as in 2011 and 2012. The reason is that the CCD model calculation is based on the benefit values of land use, which focuses more on static description; whereas the DCCD result is obtained by measuring and calculating the change rates of the land use benefits, which is more sensitive to the changes in land use benefits. We argue that our conclusion is more persuasive because we use both models.

We discovered that the result of our time series analysis on the socioeconomic and ecological environment benefits of Xiamen's land use differs from that of Mei et al. [61]. We derived that the ecological environment benefits were increasing with fluctuations and the growth rate was limited. There was even a decrease from 2008–2011. Mei's result, meanwhile, showed that the ecological environment benefits were always growing. By comparing the index systems, we found that Mei considered only the situation of pollutant control, while ignoring the production of the three types of industrial waste and household garbage. The growth of the three types of industrial waste and household garbage offset the environmental protection efforts. However, unlike other research works [62,63], we did not include some important indices such as the soil erosion rate, forest coverage rate, and sanitation conditions, due to lack of Xiamen's statistical data, which could have affected the result.

We also found, in the results, that, in early times, infrastructure construction lagged behind economic development and the discrepancy between socioeconomic and ecological environment benefits was negligible. With the growth of infrastructure, which significantly promoted the socioeconomic benefits, the turning point occurred in 2009. From then on, the socioeconomic benefits overtook the ecological environment benefits, and the gap was widening. This implies that the supply of infrastructure has significant influences on socioeconomic development. The results also show that environmental protection laws and regulations inhibited the production of the three types of waste and household sewage, which promoted the ecological environment benefits. Meanwhile, the optimization of industrial structure improved the socioeconomic benefits, as well as the ecological environment benefits, and also promoted the CCD of land use benefits. Environmental protection does not only depend on legislation and law enforcement, but also requires the advancement of science and technology. Meanwhile, the optimization of industrial structure also demands talents and technological

advancement. In the data, we found that education indices have similar weights as the infrastructure indices, which are much greater than other indices. Therefore, the government should expand the investment in education, as well as science and technology, to labor skills.

The land use in China is determined by the government. The quota of construction land is allocated by the central government to local governments, which then determines specific uses of the quota. This has indirectly restrained the rate of urbanization. When converting the arable land, other than basic arable land, to construction land, the standard of compensation is mainly decided by the government, who would take the land for a price far below the market value and use it to attract investment or for other purposes. Such a low-cost land regaining policy, on the one hand, undermines the benefits of the original user and, on the other hand, can cause inefficient land use. However, due to the unavailability of data and technical constraints, we did not consider the variations of land use benefits in such margins of urban area, hence, we could not analyze the land use benefits for each district.

## 6. Conclusions

Studying the usage coupling relationship between the socioeconomic and ecological environment benefits of land use is significantly meaningful to the harmonious development of Xiamen's urbanization and the ecological environment. The change in socioeconomic and ecological environment benefits of land use is a process that evolves dynamically. This paper leveraged the data of Xiamen City between 2002 and 2017, constructed a land use benefit system that covered four systems, i.e., social, economic, ecological, and environmental benefits, and then calculated the coupling coordinated degree among the benefits. We found that: the coupling degree of Xiamen's land use benefits is still low. The environmental protection laws, proper supply of infrastructure, and industrial optimization have positively affected the land use benefits and also promoted their coupling degree.

The above discovery has some implications regarding policy making. The city government of Xiamen should seriously consider the reality that the coupling degree of land use benefits in Xiamen is still low, and therefore decide what is to be interfered with by policies. In solving the land use problem, the market should be given a major role. The government, though, should continue perfecting the environmental protection laws and regulations, and promote the supply of infrastructure. Industrial optimization still needs the market to play a major role, and the government to play a minor role. For instance, the government could tax companies with high pollution and establish a system that monitors the pollutant handling, and by doing so, increase the cost of the polluting companies, in contrast to the government's selectively attracting some companies, which can cause insufficient competitivity of the attracted company. In the meantime, the government should expand the investment in education, as well as science and technology.

**Author Contributions:** Conceptualization, X.J. and Y.Z.; Data curation, K.W.1; Formal analysis, T.J.; Funding acquisition, KW.2; Investigation, X.J.; Methodology, X.J.; Project administration, T.J.; Software, K.W.1 and KW.2; Visualization, K.W.1; Writing—original draft, X.J.; Writing—review & editing, T.J., KW.2, and Y.Z. All authors have read and agreed to the published version of the manuscript.

**Funding:** This research was funded by the National Natural Science Foundation of China (no. 71072066), the Sichuan University (no. SKGT201602, no. 2018HHF-42), and the Department of Science and Technology of Sichuan Province (no. 2018JY0594).

**Conflicts of Interest:** All authors declare no conflict of interest.

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
