# Peer review of "Coupling Analysis of Urban Land Use Benefits: A Case Study of Xiamen City"

_land, doi:10.3390/land9050155_

Round 1

Reviewer 1 Report

This is an interesting study which utilizes long-term statistical data to construct and index system to evaluate the coupling relationship between the socioeconomic and ecological environment benefits of land use dynamics of Xiamen City in China. The study showed some interesting findings regarding coupling coordination degree of land use benefits associated with urbanization, socioeconomic and ecological environment. Some moderate modifications are necessary to render the manuscript publishable.

An important aspect of the work which authors need to cross check is the dates of the datasets which was used for the analysis in the study. Some inconsistencies exist in this regard. For instance, within the abstract, authors noted that they based their analysis on statistical data from 2002 to 2017. However, under section 3.2, authors have indicated on P4, L174-175…that quote “The data used in this study mainly comes from Yearbook of Xiamen Special Economic Zone 2013-2018, Fujian Statistical Yearbook 2003-2018 and China City Statistical Yearbook 2013-2018…unquote. Authors should cross check this inconsistency and clarify.

There are quite a number of grammatical and typographical errors which authors need to attend to. In addition to the observations highlighted above, I have provided numerous remarks and suggestions in the PDF for the attention of authors.

Author Response

Dear Reviewer:

We appreciate your detailed comments, which have greatly helped us polish our paper. We feel sorry for the defects in the previous draft submitted. We have modified our paper accordingly, as is shown in the paper with revision mode.

Specifically, we have corrected the grammatical and printing errors, and adjusted the structure of the paper. The conclusion paragraph in Introduction has been deleted and a section of discussion has been inserted between the Results and Conclusion sections. The discrepancy between the year of yearbooks and that of the data is explained in L.196-198.

Based on other reviewers’ advice, we have modified the paper as follows:

  • Added literature review regarding the development of the methodology in L.119-142;
  • Added a description of the change in land use in Xiamen City, a map showing the Xiamen’s geographical location, and a map showing Xiamen’s land use in 2017;
  • Added detailed explanation for the building of our index system in L.223-232;
  • Mentioned the pros and cons of our model and methodology in L.238-240, 252-254 and 320-325; titled every figure and table;
  • Modified the symbols and polished their definitions in 260-262, 264-267 and 281-283.

Again, we appreciate your time reviewing our paper and your comments which have been valuable to us. We would love to hear from you should there be any further questions or advice.

Best regards,

Authors

Reviewer 2 Report

Through proposing an index system covering social, economic, ecological and environmental benefits, and evaluating the CCD and DCCD of the system applied to the case of Xiamen, this paper discusses the coordination issue of the land use benefits. Generally, this paper has tried to bring some new ideas on the index selection and the application of EWM, CCD and DCCD models. However, due to the below problems, this intention is not fully realized.

  1. The structure of the paper should be improved. Firstly, the conclusion should not appear in the introduction part. Between result and conclusion, there should be a discussion part to compare this study with other similar research in terms of the construction of the index system and the development of the models.
  2. Since the construction of the index system and its evaluation are the vital and innovative part of the paper, I suggest authors focusing more on the literature review of the methodology development besides of the history of land use benefits research.
  3. I suggest authors inserting a figure in section 3.1 to illustrate the location of the case study.
  4. Please elaborate more on the construction of the index system, since this is the key part of the paper.
  5. The symbols used in several equations are misleading. For example, xi,yi in eq.(5) and (6) do not include j. Therefore, how to calculate the values of xi and yi? In addition, the symbols used in eq. (5)(6) have the same pattern as in eq.(12)(13). But eq. (12)(13) should reflect the dynamic feature.
  6. Certain important citations are missing. Therefore, it is difficult to judge the sources of the model construction either from existing literature or from authors. For instance, between L.199-202, the entropy weight method is described without citation. The same problem applies to L. 213-218, L.225-226, and L.238-271.
  7. In L.205-209, at first authors point out the shortcoming of the coupling degree, i.e. incapability of assessing the development level of individual systems. Therefore, they intend to introduce the coordination degree. In such a sense, the latter one seems dealing with individual systems. However, in L. 220-221, the text clearly describes the role of coordination index, i.e. the interaction between both socioeconomic and ecological environmental benefits. How to understand this paradox?
  8. In eq. (14), authors set up A and B to calculate the evolution state of both systems. My questions are: (1) As authors argue, “A and B can affect each other”, how can this property be reflected from the eq.? (2) This eq. in generally adopts the same pattern as T in eq. (8). What are the exact differences in the context of the introduction of dynamics? Furthermore, the significance of importing DCCD model should be addressed in the discussion part by comparing the results of CCD with DCCD.
  9. In L. 262-263, authors claim that from v=f(vA, vB), the change in v can be analyzed from controlling vA and vB. My question is how we can do this based on this general form. Can authors elaborate this process more?
  10. On the one hand, authors claim the projection is an ellipse (Fig. 1) due to the slower evolution of ecological environmental benefits than socioeconomic ones. On the other, in table 3, they state in stage 1 primary coordinated development, vA<vB. How should we understand this contradiction?
  11. All the figures and tables should include captions and the notations in certain figures include non-English characters.
  12. In L.334, based on which reason do authors introduce interval regression analysis and the separation point fall in 2010?
  13. While in eq. (20) 1<=t<=8, the sentence in L. 341 reads “t rangers from 1 to 16”. Please correct it.
  14. There is a logic gap between the results “the growth rate of socioeconomic benefits …environmental benefits” (in L.359-360) and the bullet 4 of conclusions (388-396). Please set up a linkage in the discussion part.

Author Response

Dear Reviewer:

We appreciate your detailed comments, which have greatly helped us polish our paper. We feel sorry for the defects in the previous draft submitted. We have modified our paper accordingly, as is shown in the paper with red text.

We answer your concerns as follows:

  1. The structure of the paper should be improved. Firstly, the conclusion should not appear in the introduction part. Between result and conclusion, there should be a discussion part to compare this study with other similar research in terms of the construction of the index system and the development of the models.

Per your advice, we have removed the conclusion part in the Introduction section and restructured the paper.

  1. Since the construction of the index system and its evaluation are the vital and innovative part of the paper, I suggest authors focusing more on the literature review of the methodology development besides of the history of land use benefits research.

We added literature review regarding the development of the methodology in L.118-140.

  1. I suggest authors inserting a figure in section 3.1 to illustrate the location of the case study.

We added a map showing the Xiamen’s geographical location in L.187-190, and a map showing Xiamen’s land use in 2017. A description of the change in land use in Xiamen City is also added in L.177-180.

  1. Please elaborate more on the construction of the index system, since this is the key part of the paper.

We added detailed explanation for the building of our index system in L.211-230;

  1. The symbols used in several equations are misleading. For example, xi,yi in eq.(5) and (6) do not include j. Therefore, how to calculate the values of xi and yi? In addition, the symbols used in eq. (5)(6) have the same pattern as in eq.(12)(13). But eq. (12)(13) should reflect the dynamic feature.

We modified the symbols and polished their definitions in L.258-265 and 279-281.

  1. Certain important citations are missing. Therefore, it is difficult to judge the sources of the model construction either from existing literature or from authors. For instance, between L.199-202, the entropy weight method is described without citation. The same problem applies to L. 213-218, L.225-226, and L.238-271.

We added the reference of EWM and the equation of overall benefits in L.239 and L.256. As for the equation , many research works subjectively assigned 0.5 to both a and b, while we derived their values according to our own data and calculation, which is why there was no reference. Regarding L.238-271 of the previously submitted draft (L.274-317 in this draft), you mentioned the problem of not referencing. The derivation of equations is based on the references above them, which may have resulted in an impression of having no reference.

  1. In L.205-209, at first authors point out the shortcoming of the coupling degree, i.e. incapability of assessing the development level of individual systems. Therefore, they intend to introduce the coordination degree. In such a sense, the latter one seems dealing with individual systems. However, in L. 220-221, the text clearly describes the role of coordination index, i.e. the interaction between both socioeconomic and ecological environmental benefits. How to understand this paradox?

The coupling degree indicates the mutual effects of two (or more) systems, but does not imply whether such effects are positive or negative. Two systems that are in low-level development might be highly coupled, in which case the only implication is that the mutual effects are strong. The coordination degree, on the other hand, solves this problem. It reflects both the mutual effects between systems, plus the development extent of individual systems alone.

  1. In eq. (14), authors set up A and B to calculate the evolution state of both systems. My questions are: (1) As authors argue, “A and B can affect each other”, how can this property be reflected from the eq.? (2) This eq. in generally adopts the same pattern as T in eq. (8). What are the exact differences in the context of the introduction of dynamics? Furthermore, the significance of importing DCCD model should be addressed in the discussion part by comparing the results of CCD with DCCD.

The mutual effects of A and B is explained in L.296-298 in the modified draft

We explained the symbols in L.279-281 in the modified draft

We added the discussion comparing the CCD and DCCD models in L.418-426.

  1. In L. 262-263, authors claim that from v=f(vA, vB), the change in v can be analyzed from controlling vA and vB. My question is how we can do this based on this general form. Can authors elaborate this process more?

We added explanation to this in L.301-311. The complex system is a compound of two sub-systems, so when the sub-systems are coordinated, the complex system is also coordinated. Given this, we study the projection of v in the plane , i.e., the relations between  and , to determine if the complex system is coodrdinated.

  1. On the one hand, authors claim the projection is an ellipse (Fig. 1) due to the slower evolution of ecological environmental benefits than socioeconomic ones. On the other, in table 3, they state in stage 1 primary coordinated development, vA<vB. How should we understand this contradiction?

The expression of the original text was not clear. It is corrected in l.311-313. The eventual projection is an ellipse because the maximum evolution rate of the socioeconomic system is greater than that of the ecological environment system in extreme conditions.

  1. All the figures and tables should include captions and the notations in certain figures include non-English characters.

Fixed.

  1. In L.334, based on which reason do authors introduce interval regression analysis and the separation point fall in 2010?

Added in L.383-385. We select different time points as the boundary and observe the degree of fitting, then choose the point that optimizes fitting degree on both sides.

  1. While in eq. (20) 1<=t<=8, the sentence in L. 341 reads “t rangers from 1 to 16”. Please correct it.

Done. Please see L.392-393

  1. There is a logic gap between the results “the growth rate of socioeconomic benefits …environmental benefits” (in L.359-360) and the bullet 4 of conclusions (388-396). Please set up a linkage in the discussion part.

We corrected this by rewriting the conclusion section.

Based on other reviewers’ advice, we have modified the paper as follows:

  • Rewrote the conclusion and corrected typos and grammatical problems;
  • Added a title to each figure and table;
  • Explained the inconsistency between the publication year and the actual data year (L.194-196);
  • Mentioned the pros and cons of our model and methodology in L.238-240, 252-254 and 320-325;

Again, we appreciate your time reviewing our paper and your comments which have been valuable to us. We would love to hear from you should there be any further questions or advice.

Best regards,

Authors

Reviewer 3 Report

  1. This manuscript is about the coupling analysis of urban land use benefits. Although the authors provided some findings. However, the novelty and main contribution of this study were unclear. I have given some observation below and hope these will help the authors to revise the manuscript.
  2. This journal is about "land", so I would like to suggest the authors to add and enhance the analysis between the urban land use situations and the benefits with subdivision (ex. district, county-level city..) of Xiamen City.
  3. Please make a clear definition of " Urban Land Use Benefits" in the content.
  4. This study used the Entropy Weight Method (EWM), the coupling coordination degree (CCD) model and the Dynamic coupling coordination degree (DCCD) model to evaluate the overall coupling coordination degree of land use benefits. What are the advantages and limitations of the EWM, CCD and DCCD ? Please enhance the statement in the content.
  5. Please add a clear title for every table and figure in the content. 
  6. Line 186-194, please explain how the index system of land use benefits was created ? Additionally, please add the information of reference and data source of each index.
  7. Line 195-202, please explain the data source for index weight calculate ?
  8. Please add a location map of Xiamen City in the content.
  9. Please add a urban land use map of Xiamen City in the content.
  10. Line 310 and line 330, please modify the "Chinese word" to English word for Figure 2. and Fugure3.
  11. Line 369-399, in the conclusion section, I would like to suggest the authors to add the statement about how to apply the result for urban land use management. 

Author Response

Dear Reviewer:

We appreciate your detailed comments, which have greatly helped us polish our paper. We feel sorry for the defects in the previous draft submitted. We have modified our paper accordingly, as is shown in the paper with red text.

We answer your concerns as follows:

  1. This journal is about "land", so I would like to suggest the authors to add and enhance the analysis between the urban land use situations and the benefits with subdivision (ex. district, county-level city..) of Xiamen City.

Sadly, we were unable to obtain the complete district- or county-specific data. As a result, we could not perform analysis on lower-level regions. The paper mainly discusses the land use benefits of Xiamen City. We added the induction to the land use situations in L.175-179 and leave more detailed discussion on the subdivisions may to next paper.

2.Please make a clear definition of " Urban Land Use Benefits" in the content.

Given in L.42-49.

3.This study used the Entropy Weight Method (EWM), the coupling coordination degree (CCD) model and the Dynamic coupling coordination degree (DCCD) model to evaluate the overall coupling coordination degree of land use benefits. What are the advantages and limitations of the EWM, CCD and DCCD ? Please enhance the statement in the content.

We added the the pros and cons of our model and methodology in L.238-240, 252-254 and 320-325.

4.Please add a clear title for every table and figure in the content.

Added

6.Line 195-202, please explain the data source for index weight calculate ?

We explained the building of the index system in L.211-230 and will upload the data source with the final version.

7.Please add a location map of Xiamen City in the content.

8.Please add a urban land use map of Xiamen City in the content

9.Line 310 and line 330, please modify the "Chinese word" to English word for Figure 2. and Fugure3.

10.Line 369-399, in the conclusion section, I would like to suggest the authors to add the statement about how to apply the result for urban land use management.

All fixed.

Based on other reviewers’ advice, we have further modified the paper as follows:

  • Removed the conclusion part in the introduction and restructured the paper by adding a discussion section and rewriting the conclusion;
  • Added a description of the change in land use in Xiamen City in L.177-180;
  • Modified the symbols and polished their definitions in 257-265, L.279-281;
  • Explained the inconsistency between the publication year and the actual data year (L.194-196).

Again, we appreciate your time reviewing our paper and your comments which have been valuable to us. We would love to hear from you should there be any further questions or advice.

Best regards,

Authors

Reviewer 4 Report

Dear Authors,

Minor spell check required and maybe a little bit extent of conclusions.

Author Response

Dear reviewer:

Thank you for your hard work!

We check our manuscript's spell and extent our conclusions.

Please go over the manuscript again and contact with us if you have any other suggestions.

Thank you again.

Authors

Round 2

Reviewer 2 Report

Authors have properly answered the questions I have proposed. I suggest to accept the manuscript.

Author Response

Dear reviewer:

Thank you for your hard work!

Best wishes!

Authors

Reviewer 3 Report

I appreciate how the authors responded to reviewer's comments.
This revised version has improved the manuscript clearly.
There are still two issues as follows:

1.As my original comment: This study used the Entropy Weight Method (EWM), the coupling coordination degree (CCD) model and the Dynamic coupling coordination degree (DCCD) model to evaluate the overall coupling coordination degree of land use benefits. What are the advantages and limitations of the EWM, CCD and DCCD ? Please enhance the statement in the content.

The authors answer: We added the the pros and cons of our model and methodology in L.238-240, 252-254 and 320-325.
However, what are the advantages and limitations are still to be doubted.

2.Line 252-253, according to the "Table 1.The evaluation index system of land use benefits.", the authors should explain how and why the 21 secondary indices were chosen. Additionally, are there references to support them, please cite it for every index.

Author Response

Dear Reviewer:

We do appreciate your dedication to the details, which has helped us with further polishing the paper. We are sorry for the problems that remained in the modified draft. We have solved them according to your comments.

1.As my original comment: This study used the Entropy Weight Method (EWM), the coupling coordination degree (CCD) model and the Dynamic coupling coordination degree (DCCD) model to evaluate the overall coupling coordination degree of land use benefits. What are the advantages and limitations of the EWM, CCD and DCCD ? Please enhance the statement in the content.

The authors answer: We added the the pros and cons of our model and methodology in L.238-240, 252-254 and 320-325. 
However, what are the advantages and limitations are still to be doubted.

We added the pros and cons of various methods in L.244-250,L284-292,L340-354 and explained why we used them.

2.Line 252-253, according to the "Table 1.The evaluation index system of land use benefits.", the authors should explain how and why the 21 secondary indices were chosen. Additionally, are there references to support them, please cite it for every index.

We rewrote the construction of the index system in L.207-239, including why researchers used to choose them and the problems that we faced in our research.

In addition, we also corrected some detailed errors:

  • Added the definition of the variables in L.253,L255,L257;
  • Changed the interval of t in L.424.

Again, we are grateful for your comments and we are happy to answer your further questions.

Best Regards,

Kun Wang